# Heartwood Extract of *Pterocarpus marsupium* Roxb. Offers Defense against Oxyradicals and Improves Glucose Uptake in HepG2 Cells

**DOI:** 10.3390/metabo12100947

**Published:** 2022-10-05

**Authors:** Mohammad Irfan Dar, Sahar Rafat, Kapil Dev, Sageer Abass, Mohammad Umar Khan, Walaa A. Abualsunun, Samar S. Murshid, Sayeed Ahmad, Mohammad Irfan Qureshi

**Affiliations:** 1Department of Biotechnology, Jamia Millia Islamia, New Delhi 110025, India; 2Bioactive Natural Product Laboratory, School of Pharmaceutical Education and Research, Jamia Hamdard, New Delhi 110062, India; 3Department of Food Technology School of Interdisciplinary Science & Technology, Jamia Hamdard, New Delhi 110062, India; 4Department of Pharmaceutics, Faculty of Pharmacy, King Abdulaziz University, Jeddah 21589, Saudi Arabia; 5Department of Natural Products and Alternative Medicine, Faculty of Pharmacy, King Abdulaziz University, Jeddah 21589, Saudi Arabia

**Keywords:** health care, diabetes, *Pterocarpus marsupium*, UP-LCMS, HepG2 cells, oxidative stress

## Abstract

Diabetes mellitus leads to cellular damage and causes apoptosis by oxidative stress. Heartwood extract of *Pterocarpus marsupium* has been used in Ayurveda to treat various diseases such as leprosy, diabetes, asthma, and bronchitis. In this study, we worked out the mechanism of the antidiabetic potential of methanolic heartwood extract of *Pterocarpus marsupium* (MPME). First, metabolic profiling of MPME was done using gas chromatography-mass spectrometry (GCMS), ultra-performance liquid chromatography-mass spectroscopy (UPLC-MS), and high-performance thin-layer chromatography (HPTLC) to identify phenols, flavonoids, and terpenoids in MPME. Biological studies were carried out in vitro using the HepG2 cell line. Many antidiabetic compounds were identified including Quercetin. Methanolic extract of MPME (23.43 µg/mL–93.75 µg/mL) was found to be safe and effective in reducing oxyradicals in HepG2 cells. A concentration of 93.75 µg/mL improved glucose uptake efficiently. A significant decrease in oxidative stress, cell damage, and apoptosis was found in MPME-treated HepG2 cells. The study suggests that the heartwood of *Pterocarpus marsupium* offers good defense in HepG2 cells against oxidative stress and improves glucose uptake. The results show the significant antidiabetic potential of MPME using a HepG2 cell model. The effect seems to occur by reducing oxidative stress and sensitizing the cells towards glucose uptake, hence lowering systemic glucose levels, as well as rescuing ROS generation.

## 1. Introduction

Diabetes mellitus (DM) is a term used to describe a group of metabolic abnormalities marked by persistent hyperglycemia. Type 2 diabetes mellitus (T2DM) is sometimes cited as non-insulin-dependent diabetes mellitus (NIDDM) as it occurs because of the inability of beta cells of the pancreas to secrete insulin, defects in insulin action, or both [1]. T2DM is linked to acute and chronic complications, which cause most DM-related fatalities, as well as financial strain and poor quality of life. Diabetes prevalence in the 20 to 79-year age group was estimated to be 463 million in 2019, and this number is anticipated to rise to 700 million by 2045 [2]. Diabetes alone is responsible for about 1 million deaths per year, making it the tenth-largest cause of human death [3]. Oxidative stress is significantly linked to diabetes mellitus (DM) [4]. Glucose-autooxidation and protein metabolism leads to excessive production of reactive oxygen species (ROS), and chronic hyperglycemia caused by diabetes increases oxidative stress and damages the structural elements of cells. It adversely affects nerves, blood vessels, and delicate organs, including the eyes and kidneys [5].

Patients with DM are more likely to develop significant complications such as dyslipidemia and cardiovascular disease, which are the leading causes of fatality. Diabetes has an impact on contagious diseases, and post-disease side effects, and can cause fatality from severe infections, increasing the risk of complications along with comorbidities in patients due to the coronavirus outburst (COVID-19), which has evolved into a global health crisis [6]. There is a need for proper medicine to counter diabetes. Antidiabetic drugs based on modern allopathy are time-consuming, need to be taken in multiple doses and are frequently taken by patients for the rest of their lives. For the treatment of diabetes, a variety of chemical hypoglycemic drugs such as metformin, acarbose are meglitinides are available. However, side effects such as increased cardiovascular risk, gastrointestinal problems, stomach discomfort, diarrhea, and hepatotoxicity have been linked to these synthetic medications [7]. As a result, natural products are in great demand for the treatment of diabetes and its complications because they have fewer side effects than synthetic drugs. The ability of phytochemicals to function as substitutes for allopathic medications used in diabetes treatment involves their strong antioxidant capability, which aids in battling oxidative stress as well as other cellular injuries, and provides new insight into treating diabetes mellitus [8]. Some medicinal plants are reported to regulate different pathways that control blood glucose and activate fatty acid oxidation linked to insulin resistance [9]. Medicinal herbs are precursors for producing valuable medications because of their safety, cheap cost, availability, and manageable side effects.

The deciduous tree *Pterocarpus marsupium Roxb.* (PM) is found in Sri Lanka and India. It is known in Hindi as Vijaysar and is a significant medicinal plant used as a folk medicine to treat diabetes. Because of its therapeutic and laxative effects, it is well-known in the Ayurvedic system of medicine. The heartwood of PM is used as a depurative, hemostatic, and rejuvenating agent, and is used to treat many life-threatening diseases such as diabetes, bronchitis, and leprosy [10]. The bark, seed, and heartwood of PM have been particularly beneficial in decreasing hyperglycemia in pharmacological trials [11]. Despite carrying a rich content of valuable phytochemicals, minimal attention has been paid to exploring the anti-diabetic properties of this plant. In this study, we studied the possible anti-diabetic mechanism of *Pterocarpus marsupium*. The outcome of the study may result in managing diabetes with fewer side effects at a low cost. In this study, we used an in vitro HepG2 cell model to evaluate the therapeutic potential of a methanolic heartwood extract of *Pterocarpus marsupium* (MPME).

To the best of our knowledge, there have been no reports on the antidiabetic effect of the heartwood of PM on HepG2 cells. The study shows the chemoprotective effect of heartwood extract of PM against ROS stimulated by high glucose stress in HepG2 cultured cells. High-throughput screening, cellular antioxidant potential, and glucose uptake potential were explored to provide specific insights into the mechanism of action of PM against type 2 diabetes mellitus.

## 2. Materials and Methods

Materials were acquired as follows: TLC silica gel 60F254 (Merck KGaA Darmstadt, Germany), Dulbecco’s Modified Eagle’s Medium (DMEM), fetal bovine serum (FBS) (GIBCO, Waltham, MA, USA), 3-(4,5-dimethylthiazol-2-yl)2,5-diphenyl tetrazolium bromide (MTT), trypsin, propidium iodide (Himedia, India), glucose uptake-assay-kit (Eton Bioscience, SKU-1200032002), acridine orange (Merck, Germany), ethidium bromide (Invitrogen, Waltham, MA, USA), 4′,6-diamidino-2-phenylindole (DAPI) (Thermo Fisher, Waltham, MA, USA), and antibiotic (penicillin-streptomycin) (CAISSON, Smithfield, UT, USA). Unless otherwise specified, all cell culture chemicals and standards were purchased from Sigma (St. Louis, MI, USA).

### 2.1. Collection of Plant Material

The selected plant material was procured and authenticated as per the standard protocol specified in the Ayurvedic Pharmacopeia of India. The authenticated plant material was deposited in the Bioactive Natural Product Laboratory for future reference with voucher no JH/BNPL/SA/2020/PM.

### 2.2. Preparation of Plant Extracts

Well-grown and healthy heartwood of *Pterocarpus marsupium* was rinsed with distilled water and dried in a hot-air oven at 40 °C. The dried plant part was ground into a fine powder, and 50 g of plant sample was extracted in 500 mL of methanol at ambient temperature for 5 h using a Soxhlet device. After cooling, the extract was filtered, and the filtrate was dried by incubation at 35–38 °C at reduced pressure. Further, the percentage yield of the obtained crude methanolic extract was calculated and then stored at 4 °C for further experiments.
%Yield of extract (g/100 g) = Weight of dried extract × 100/Weight of plant material

### 2.3. Physicochemical Analysis

The powdered sample was tested for physicochemical parameters including total ash, acid insoluble ash, and water-soluble ash, as well as moisture content, according to WHO guidelines on quality control methods for medicinal plant materials and standard methods prescribed in the Indian Pharmacopeia [12].

### 2.4. Estimation of Total Phenolic and Flavonoid Content

The Folin Ciocalteu method was employed to determine the total phenol content (TPC) of MPME [13]. Gallic acid was used as a standard. A UV spectrophotometer (*i*Mark Microplate Reader Bio-Rad, Hercules, CA, USA) was used to measure optical density at 765 nm. The standard curve of gallic acid (10−1000 μg/mL) was used to derive the estimation of TFC of MPME and the content was expressed as mg gallic acid equivalent/gm of MPME (mg GAE/gm of MPME). The aluminum chloride colorimetric method was employed to determine the total flavonoid content (TFC) of the MPME [14] and read at 415 nm. TFC was calculated from a standard curve of rutin (10−1000 μg/mL), and the content was expressed as mg rutin equivalent/gm MPME (mg rutin/gm of MPME).

### 2.5. Phytochemical Profiling by GCMS

Precisely 500 μL of the sample was mixed with 500 μL of hexane into a centrifuge tube and then passed through a 0.25 μ polytetrafluoroethylene membrane filter. A 2 μL filtered sample was processed in a GC–MS (Agilent 7890A series, Agilent, Santa Clara, CA, USA) with CTC-PAL autosampler attached to HP-5MS (5% phenyl polydimethylsiloxane) capillary column (30 m × 0.25 mm I.D and 0.25 μm film thickness) and associated with a mass detector (Agilent 5975C inert XL EI/CI MSD with Triple-Axis Detector, Agilent Technology, Santa Clara, CA, USA). The temperature of the instrument was set at 50 °C for 5 min and was then elevated up to 280 °C at a specific rate of 10 °C/min, and for 7 min with a run time of 42 min. Mass spectra were reported from m/z 50 to 700 at 0.5 s/scan. The software installed within the instrument, Wiley and NIST (comparison software National Institute of Standards and Technology) library, was involved in the identification of phytoconstituents [15].

### 2.6. HPTLC Fingerprinting and Quantitative Analysis

In a stationary phase, an aluminum silica gel 60 F_254_ plates (20 × 10 cm, 0.2 mm thickness) was employed on which the sample and standard Quercetin were adsorbed using a CAMMAG Linomat V at a pace of 150 nL/s, and in the arrangement of bands (6.0 mm). The mobile phase for TLC development was made up of toluene, ethyl acetate, methanol, and formic acid (6:3:0.5:0.5, *v*/*v*/*v*/*v*). TLC was developed in a twin trough glass chamber after saturation of the mobile phase for 30 min at room temperature up to a length of 80 mm. After drying by an air-dryer, the TLC plate was scanned and digitized. Densitometric scanning was carried out on a CAMAG TLC Scanner 3 at absorbances of 254 nm and 366 nm. The concentration of Quercetin was measured quantitatively at 254 nm [16].

### 2.7. Ultraperformance Liquid Chromatography-Mass Spectroscopy (UPLC-MS) Studies

UPLC-MS analysis of MPME was performed using a Waters ACQUITY UPLC system (Serial No. #F09 UPB 920M; Model code # UPB; Waters Corp., Milford, MA, USA) equipped with a binary solvent delivery system, column manager, autosampler, and a tunable MS detector (Serial No # JAA 272; Synapt; Waters, Manchester, UK). MPME was chromatographically separated in gradient elution mode using a previously degassed mobile phase containing 0.3% *v*/*v* formic acid in water (A) and acetonitrile (B) (initially, 10% A; 0–5 min 40% A; 5–10 min 60% A; 10–13 min, 90% A; 13–15 min, 100% A, 15–16 min 10% A).The total run time was 16 min. The mobile phase flow rate was 0.4 mL/min and the column used was Water’s ACQUITYUPLCBEH C18 (100 2.1 mm 1.7 m). Precisely 10 µL of the sample was injected with the help of an auto-injector, in a split mode of 5:1, and the system pressure was set to 15,000 psi. The MS detector identified the separated metabolites. The UPLC and mass detector were both regulated by MassLynxV4.1 software installed with this instrument. A literature search was used to identify the separated compounds based on their m/z value [15].

### 2.8. Cell Culture and Treatments

The HepG2 cell line was gifted by Dr. Sayed Naqui Kazim Jamia Millia Islamia. Stored cells were revived and cultured in Dulbecco’s Modified Eagle Medium (DMEM) supplemented with 10% (*v*/*v*) fetal bovine serum (FBS), penicillin (100 IU/mL), and streptomycin (100 µg/mL) in a humidified atmosphere at 5% CO_2_ at 37 °C until confluent. Confluent cells were sub-cultured every three days at 70–80 percent confluence using a trypsin/EDTA solution.

#### 2.8.1. MTT {3-(4,5-Dimethylthiazol-2-yl)-2,5-diphenyltetrazolium bromide} Assay

The cytotoxicity of MPME was measured by an MTT assay following a standard protocol [17]. In brief, 1 × 104 cells/well in a 96-well plate were treated with successive dilutions of extract (1500, 750, 375, 187.5, 93.75, 46.87, and 23.43 µg/mL) in complete DMEM and incubated for 24 h. Following incubation at 37 °C, a further 20 µL of MTT solution (5 mg/mL) was poured into each well and incubated for three hours. After incubation, the supernatant was discarded from each well, and formazan crystals were solubilized with 100 μL of dimethyl sulfoxide (DMSO) added to solubilize formazan crystals. The absorbance was recorded at 570 nm. The following equation was used to calculate data from three different experiments.
Cell viability (%) = (OD sample value)/(OD cell control) × 100.

#### 2.8.2. Measurement of Glucose-Uptake

Cells were cultured in 12-well plates and incubated for 24 h, at 37 °C in a CO_2_ incubator. After 24 h, the media was discarded, and cells were washed with PBS. Cells were treated with MPME in serum-free DMEM containing 0.2% BSA and incubated for a further 24 h at 37 °C in a CO_2_ incubator. Glucose uptake in test samples and controls was measured in the spent media using a kit (Eton Bioscience, SKU-1200032002) as per the manufacturer’s instructions.

#### 2.8.3. Detection of ROS in HepG2 Cells by Flow Cytometry Using 2′,7′-Dichlorodihydrofluorescein Diacetate (DCFDA)

The redox-sensitive fluorescence probe carboxy-2′,7′-dichlorodihydrofluorescein di-acetate carboxy-H2DCFDA; was used to estimate ROS production as per a previously reported method [18]. Briefly, HepG2 cells (2 × 10^5^) were grown in 6-well plates and treated with different concentrations (23.43, 46.87, and 93.75 µg/mL) of MPME-containing media for 24. Media was removed, and cells were washed with PBS before inducing oxidative stress using 100 μM H_2_O_2_ for 1 h. The cells were trypsinized, 10 μM DCFH-DA was added to the culture media, and cells were incubated in the dark for 30 min. After incubation, the cells were washed with serum-free media to remove the unbound dye, followed by re-suspending the cells in PBS to measure ROS employing flow cytometry (BD FACS Aria III, USA) with an excitation wavelength of 485 nm and an emission wavelength of 525 nm.

#### 2.8.4. Assessment of Apoptosis and Necrosis

HepG2 cells grown in 6-well plates were incubated with DMEM media with or without high glucose (HG) (50 mM) for 24 h at 37 °C. After incubation, MPME at varying concentrations was added to respective wells. After incubation, the extract was removed and cells were washed with PBS and 25 µL (1 × 10^5^) of HepG2 cells were stained in a microcentrifuge tube with 5 µL of AO-EtBr (acridine orange (AO, 50 μg/mL) a green-fluorescent dye and ethidium bromide (EtBr, 100 μg/mL) a red fluorescent dye, for 15 min at 37 °C and placed in the dark with gentle mixing. A volume of 10 µL of cell suspension was mounted on a glass slide, covered with a glass coverslip, and examined with a fluorescence microscope with a fluorescein filter. Dual staining was visualized under a fluorescent microscope (Zoe fluorescent cell imager Bio-Rad, USA) with excitation at 488 nm and 550 nm [19].

#### 2.8.5. Morphological Changes in HepG2 Cells

To visualize morphological changes in HepG2 cell nuclei, the cells were stained with DAPI and PI dyes and visualized using a fluorescence/confocal microscope.

##### Fluorescence Microscopy

Propidium iodide is a red fluorescent dye, only permeable when the cell structure is destabilized. It passes through the cell nucleus, intercalates between the bases of DNA, and shows fluorescence with an excitation/emission-maxima ~535/617 nm. The cells were incubated with or without HG (50 mM) and then treated with MPME (93.75 μg/mL) in selected wells. After incubation at 37 °C, the cells were washed with PBS and stained with PI (2.5 ng/μg) solution and incubated in the dark for 15 min at 25 °C [20]. The cells were visualized under a fluorescent microscope (Zoe fluorescent cell imager, Bio-Rad, USA) using a TRITC filter. Images were taken at 20× magnification.

##### Confocal Microscopy

Briefly, the HepG2 cells (1 × 10^6^ cells/well) were seeded on 12-well plates [21]. After a 24-h pre-incubation period with or without glucose (50 mM), MPME (93.75 μg/mL) was added to selected wells. After 48 h of incubation at 37 °C, the cells were rinsed with PBS and fixed with 4% paraformaldehyde followed by 70% ethanol. The fixed cells were washed and stained with 100 μL (1 μg/mL) DAPI dye before being incubated at 37 °C for 30 min in a dark room. The unbound dye was then removed and 20 μL of PBS:glycerine (1:1) was added to the cells. Apoptotic cells, characterized by morphological changes in apoptotic nuclei, were observed using confocal microscopy (TCS SPS, Leica, Germany).

### 2.9. Statistical Analysis

Graph Pad Prism software Version 5.01 was used for statistical analysis. Comparison between groups was performed using Student’s *t*-test or one-way ANOVA and Tukey’s multiple comparison tests. Difference were considered statistically significant at *p* ˂ 0.05.

## 3. Results

### 3.1. Physicochemical Analysis

The outcomes of physicochemical tests were obtained within the allowed limits (Appendix A). The percentage yield of the MPME extract was 11.125 ± 0.95% (*w*/*w*). The moisture content percentage was 1.25%. Low moisture content helped to increase the shelf life of MPME.

### 3.2. Total Phenolic and Flavonoid Content of the Extract

Using a gallic acid standard calibration curve (y = 0.0049x + 0.0597: R = 0.9969), the total phenolic content of the MPME was determined and was expressed as milligram equivalents of gallic acid per gram of dry extract (mg GAE/g). The total phenolic content was 157.3 ± 10.219 mg GAE/g of MPME (n = 3). The total flavonoid content was expressed using a standard calibration curve of rutin (y = 0.0052x + 0.0049: R = 0.9921) and the result was expressed as milligram equivalents of Rutin per gram of dry extract. The MPME contained a total flavonoid content of 78.121 ± 3.190 mg rutin/g of MPME (n = 3). Pterocarpus species are rich in phytoconstituents, including flavonoids, phenolic compounds, and terpenoids [22]. The high phenolic content of this plant is associated with significant health-promoting effects.

### 3.3. Gas Chromatography-Mass Spectrometry (GC-MS) Analysis

The GC–MS chromatogram of MPME showed a total of 10 major peaks representing active compounds that were identified by peak retention time, peak area (%), height (%), and spectral fragmentation patterns of their mass to that of known compounds described by the National Institute of Standards and Technology (NIST) library (Appendix A). The tentatively recognized phytoconstituents are given in Table 1. Many antidiabetic constituents were identified such as myristic acid (18.095 Rt and 0.80% peak area), succinic acid (10.153 Rtand 0.76% peak area), (18.095 Rtand 0.80% peak area), isoeugenol (14.691 Rt and 0.80% peak area), pentadecanoic acid (19.658 Rt and 0.36% peak area), oleic Acid (30.449 Rtand 0.19% peak area), citronellal (20.361 Rtand 0.36% peak area), pentatriacontane (19.846 Rtand 0.02% peak area), and 14b-pregnane (24.064 Rtand 0.02% peak area)

### 3.4. HPTLC Fingerprinting and Quantitative Analysis

The current study showed many major and minor metabolites indicated by their retention factors (Appendix A). The validation analysis for Quercetin was shown to be linear, precise, and consistent at a concentration range of 200–4000 ng/spot. For standard Quercetin, the calibration equation and coefficient were determined to be 1.066 + 458.5, and 0.993, respectively. For Quercetin, the limits of detection (LOD) and quantitation (LOQ) were found to be 13.65, and 41.39 ng/spot, respectively. Intra-day and inter-day precision were calculated as percent (%) relative standard deviation or the coefficient of variation, with values ranging from 0.44–0.79 and 1.44–1.66. The percentage of drug recovery was 99.21–102.66% (Appendix A). In addition, the concentration of Quercetin in MPME was determined and found to be 0.409 ± 1.34 mg/g of the sample. Figure 1 shows chromatograms of the TLC plate at various wavelengths. Active constituents present in MPME may contribute to the robust antioxidant and antidiabetic activity of the plant.

### 3.5. Ultra-Performance Liquid Chromatography-Mass Spectroscopy (UPLC-MS) Studies

Metabolic profiling of MPME was determined using ultra-performance liquid chromatography-mass spectroscopy. MPME yielded a total of 29 major metabolites, (Table 2). The study was performed in a positive mode to identify the metabolites present in MPME. A chromatogram of the analysis is shown in Appendix A. The presence of major anti-diabetic bioactive compounds detected using UPLC-MS-based metabolite profiling included epicatechin (Rt 3.047), kaempferol-7-O-alpha-L-rhamnoside (Rt 3.047), curcumol (Rt 4.000), quercetin (Rt 4.000), iso-liquiritigenin (Rt 4.000), trans-pterostilbene (Rt 4.00), berberine (Rt 4.00), pinocembrin (Rt 4.000), formononetin (Rt 4.374), and amygdalin (Rt 6.519).

### 3.6. Studies Using the HepG2 Cell Line

#### 3.6.1. Cytotoxic Effect of MPME on HepG2 Cell Line

The evaluation of cytotoxicity is regarded as a crucial step in estimating potential for the use of phytochemicals as drugs [23]. The MTT cell viability assay, which depends on the mitochondrial metabolic activity of viable cells, was used to investigate MPME on an HepG2 cell line with an MPME from a concentration of 1500 μg/mL to 23.43 μg/mL. The results of cell viability were revealed after 24 h of incubation of HepG2 cells with MPME, (Figure 2B). Higher concentrations (1500 μg/mL–750 μg/mL) of MPME-treated cells showed altered morphology, decrease in cell number, and shrinkage. The control cells were well attached and exhibited normal morphology (Figure 2A).

#### 3.6.2. Glucose Uptake Assay

The effects of different concentrations (1500 µg/mL–23.43 µg/mL) of MPME on cell viability were investigated. Cell viability was initially evaluated to identify a safe dose with low cell cytotoxicity. This was further used to determine a safe dosage for the glucose uptake assay in HepG2 cells. A concentration of 96.75 μg/mL of MPME revealed significant cell viability, and this concentration of MPME was used for the analysis of glucose uptake in HepG2 cells. Excessive yellowing media was linked with high glucose utilization, and significant glucose uptake was observed in metformin-treated (positive control) cells. A significant increase in the glucose uptake by HepG2 cells was also observed after treatment with MPME (93.75 µg/mL) in comparison with the normal control (Figure 2C).

#### 3.6.3. Detection of ROS in H_2_O_2_ Induced HepG2 Cells Using 2′,7′-Dichlorodihydrofluorescein Diacetate (DCFDA) Probe by Flow Cytometry

Measurement of ROS production is a clear sign of free radical-induced cell damage observed by non-fluorescent dye DCFH-DA. The ability of MPME to rescue H_2_O_2_-induced ROS production in HepG2 cells was quantified by measuring the fluorescence intensity of reacting DCFH and oxidative substrates. The fluorescent intensity of induced cells significantly increased when compared to untreated control cells. MPME (23.43, 46.87, and 93.75 μg/mL) retained high cell viability, and when added to H_2_O_2_-induced HepG2 cells, a significant reduction in fluorescent intensity was observed in a dose-dependent manner compared to untreated (control) cells (Figure 3).

#### 3.6.4. Assessment of Apoptosis and Necrosis by Fluorescence Microscopy

HepG2 cells labeled with AO/EB after incubation in high glucose (HG) media showed much less apoptosis after treatment with MPME (93.75 μg/mL), and a significant protective effect of MPME was observed (Figure 4e). Late-stage apoptotic cells were seen with dense and abnormally localized orange nuclear EB labeling in untreated and HG-induced HepG2 cells (Figure 4b). The frequency of early-stage apoptotic and necrotic cells increased with a decrease in concentration (46.87 and 23.43 μg/mL) of MPME in HG-induced cells (Figure 4c,d). Untreated and induced cells as control showed healthy cells with the least apoptosis (Figure 4a). It was observed that MPME reduced late apoptosis and necrosis of HG-induced HepG2 cells in a concentration-dependent manner.

#### 3.6.5. Assessment of Cellular Morphology on High Glucose (50 mM)-Induced HepG2 Cells

HG-induced oxidative stress caused cell damage and apoptosis in HepG2 cells, as observed by DAPI (4′,6-diamidino-2-phenylindole) and PI (Propidium iodide). It was observed that MPME (93.75 μg/mL) treatment reduced cell damage and apoptosis in HG-induced HepG2 cells when compared with untreated and HG-induced cells with different degrees of nuclear fragmentation and chromatin condensation, after PI staining as indicated by yellow arrows (Figure 5A–C). DAPI staining was performed to observe the protective effect of MPME (93.75 μg/mL) in high glucose-induced oxidative stress in HepG2 cells. The apoptotic cells appeared bright white, deformed, and florescent, while the normal viable cells appeared deep blue and evenly distributed and with uniform shape with DAPI staining (Figure 5E–G).

## 4. Discussion

Diabetes mellitus is one of the primary causes of mortality worldwide, and its prevalence is expected to increase in the coming decades. Studies have shown that diabetics are more susceptible to certain infectious illnesses, including COVID-19, resulting in serious complications [24]. Intensive research is going on to control and treat diabetes by targeting glucose upregulation, antioxidant activity, β cell regeneration, and several other combined mechanisms [25].

In this study, the methanolic heartwood extract of *Pterocarpus marsupium* (MPME) was found to have high phenolic and flavonoid content and is expected to play significant therapeutic roles. In this study, we identified many known antidiabetic saturated and unsaturated fatty acids, alcohols, and sterols such as myristic acid, succinic acid, isoeugenol, linolic acid, and citronellal, in MPME. Isoeugenol has been reported to activate AMP-activated protein kinase (AMPK) and improved glucose uptake in rat L6 myotubes [26]. Succinic acid has been used in insulin secretagogues, and its prolonged use improved pancreatic beta cell functioning [23]. The presence of these compounds in MPME may improve its antidiabetic potential. HPTLC is a linear, precise, and accurate approach for identifying herbal constituents [27]. The mobile phase showed sharp and defined peaks seen at different Rf values, at wavelengths of 245 nm and 366 nm, revealing the presence of various plant metabolites. MPME also had a significant amount of Quercetin, at 254 nm.

The presence of Quercetin in PM, using LC-MS analysis, has been reported earlier [28]. The robust antidiabetic property of Quercetin has also been reported [29]. The presence of Quercetin in MPME may promote its strong antidiabetic potential by decreasing oxyradicals and by enhancing antioxidant enzyme activity. Furthermore, diverse polar and nonpolar metabolites were present as identified using UPLC-MS.UPLC-MS analysis of MPME, which identified the presence of various potent antidiabetic and antioxidant metabolites such as epicatechin (Rt 3.047), quercetin (Rt 4.000), curcumol (Rt 4.000), trans-pterostilbene (Rt 4.000), and formononetin (Rt 4.374). Epicatechin (Rt 3.047) is a monomeric flavan-3-ol that is recognized for its antioxidant properties against free radicals. It has been isolated from the bark of *P. marsupium* and was shown to play a functional role in reducing oxidative and antidiabetic complications in both in vitro and in vivo studies [30]. Curcumol (Rt 4.000) has been shown to reduce acute and subchronic hyperglycemia [31]. Quercetin (Rt 4.000) is a flavonoid in the bark of *P. marsupium* and has also been reported in berries and citrus fruits. Quercetin is known for its antioxidant activity and has been shown to alleviate diabetes in rats [32]. Formononetin (Rt 4.374) is an O-methylated isoflavone and has already been identified in plants [28].

Formononetin has antioxidant properties and has been shown to attenuate kidney damage in type-2 diabetic rats [33]. Trans-pterostilbene (Rt 4.000) is a methoxybenzene and a diether, and has been shown to have an anti-diabetic effect via the PI3K/Akt signaling pathway in diabetic rats [33]. The presence of these antioxidant and antidiabetic compound in MPME may be the basis for its therapeutic properties, since it is rich in phytoconstituents including flavonoids [34], phenolic compounds, and terpenoids [22]. Recent studies have confirmed the major role of free radicles in life-threatening diseases including cancer and diabetes [35]. A HepG2 cell model was previously used to study the effects of various drugs on hyperglycemia [36]. In the present study, the antidiabetic property of MPME was investigated on HepG2 cells, and this study of MPME is the first report on its used with the HepG2 cell line. An MTT-based standard method was used to assess the cytotoxicity of MPME on the HepG2 cell line, and HepG2 cells were treated with different concentrations of MPME. Interestingly MPME was less toxic to the HepG2 cell line, which retained more than 75% cell viability at a concentration <100 μg mL. The concentration at which the MPME caused a 50% reduction in cells viability (IC_50_) was 552 μg/mL, which concurs with the previous experiments that reported moderate toxicity of the plant extract on other cancer cell lines, such as HeLa and A549 [37]. MPME showed less toxicity to HepG2 cells, while other species of the same genus, showed high toxicity to other cancer cell lines. For example, a methanolic extract of *P. santalinus* showed high toxicity to HeLa cells with an IC_50_ value of 40 μg/mL. Moreover, the bioactive compounds of *P. santalinus* also showed high toxicity on HepG2 cells [38].

Based on these results, <100 μg/mL concentrations of MPME were selected as a safe dose for conducting further experiments. Induction of glucose uptake and insulin-mimetic ability has been shown by many medicinal plants. This was attributed to the presence of various phytoconstituents that help in reducing blood glucose levels and alleviating diabetic complications [39]. The HepG2 cell line was earlier used as a model to study glucose uptake measurements [40]. In previous studies, it was reported that phytoconstituent from heartwood extract of MPME improved glucose uptake in different cell models [41]. The reason for high glucose uptake in the presence of MPME was attributed to the high content of phenolic compounds, tannins, and flavonoids present in MPME, as reported earlier [40]. MPME was previously reported to increase glucose uptake up to three-fold in MCF-7 cells [42]. The possible mechanism of glucose uptake by MPME-treated cells may be the engagement of antidiabetic phytoconstituents with a protein factor that may be involved in the insulin-mediated glucose transport signaling pathway, resulting in glucose uptake activation.

Glucose uptake is regulated by the activation of AMP-activated protein kinase (AMPK) via an insulin-independent mechanism. AMPK, and Protein kinase B (Akt) phosphorylation is supposed to be activated by the high phenolic content [43,44]. Flavonoids present in the extract may also be responsible for the increase in glucose uptake [43]. The current work, which employed the HepG2 cell model, aspired to study the protective capability of MPME from oxidative stress that leads to toxicity in the liver. Hyperglycemia leads to the production of reactive oxygen species (ROS), thereby resulting in DNA damage and causing apoptosis. MPME treatment may enhance the activities of antioxidant enzymes and significantly reduce intracellular ROS in induced HepG2 cells.. Previous studies have reported an increase in oxidative enzymes after treatment with heartwood extract of *P. marsupium* in MCF-7 cells [42].

A previous study reported that Pterostilbene, one of the major constituents of MPME, protects hippocampus neuronal cells from induced oxidative damage [45]. The ROS scavenging potential of MPME removes free radicles that cause cell damage, and this potential of MPME may help diabetic patients reduce diabetic complications. High glucose concentrations (HG) (50 mM) have been utilized to investigate hyperglycemia-induced oxidative damage in numerous tissues, as well as to precisely mimic the in vivo diabetic ketoacidosis observed in patients with acute or uncontrolled diabetes [46]. HepG2 cells undergo oxidative stress in the HG environment and show greater frequency of cell damage and apoptosis [47]. The current study investigated the therapeutic potential of MPME in a high glucose environment. Dual (AO/EB) staining showed an increased level of late apoptosis and necrosis due to increased oxidative stress in HG (50 mM)-induced cells. Different treatments with MPME showed a significant dose-dependent protective effect with low apoptosis in comparison to the negative control. It should be noted that necrosis is caused by cell injury due to high oxidative stress. Therefore, MPME works by rescuing cells from glucose-induced oxidative stress. Cell damage was evident in glucose-induced HepG2 cells with compromised cell membranes that allowed propidium iodide (PI) dye to bind within the nucleus. MPME-treated cells showed significantly less damage in comparison to the negative control.

The protective effect from HG-induced HepG2 cells offered by PMEM was also confirmed by DAPI nuclear staining. It was observed that HG-induced oxidative stress damaged the nucleus of HepG2 cells, and the healing effect of MPME in HG-induced cells was observed after DAPI staining by confocal microscopy. The literature suggests a rise in ROS production in induced HepG2 cells as one of the possible causes of cell membrane degradation and nucleus disintegration. Our results show the protective effect of MPME in HG-induced oxidative stress in HepG2 cells. The presence of various, identified antioxidant phytoconstituents, and their quenching potential, might be responsible for oxyradical detoxification and protecting cells from oxidative damage. A schematic representation of the possible mechanism of antidiabetic and cellular antioxidant activity offered by bioactive compounds identified by high-throughput analysis of MPME is shown in Figure 6.

From the present study, PM used as a traditional folk medicine is a potential hepatoprotective, shows ROS scavenging activity, and alleviates HG-induced toxicity via its antioxidant activities and high glucose uptake. The phytochemicals found in MPME may have an efficacious and protective effect against diabetes, which could be linked to both direct and indirect effects. MPME may help patients with comorbidities and exposure to pathogenic viruses such as SARS-CoV-2 that causes COVID-19. Furthermore, natural and herbal medicines have very less side effects and are therefore safer as well as being more effective in a holistic medicine. The metabolites identified in MPME have future potential as antidiabetic herbal drugs. Further exploration of the mechanism of action at the level of diabetes control would be of great therapeutic interest.

## 5. Conclusions

The present study detected and identified more than 20 antidiabetic compounds from MPME using UP-LCMS and GCMS analysis. The antidiabetic and antioxidant compound Quercetin was estimated quantitatively by HPTLC and its presence may contribute to the therapeutic potential of MPME. This study demonstrated the antidiabetic potential of MPME. The possible mechanism seems to be via sensitizing HepG2 cells towards glucose uptake. Cellular ROS scavenging activity showed by MPME in HepG2 cells may be due to an increase in the activity of the enzymes responsible for antioxidant potential, thereby inhibiting cellular damage and cell death. The study also demonstrated the hepatoprotective ability of MPME against high glucose (50 mM). The therapeutic potential for rescuing cells from high glucose-induced apoptosis may result from its high antioxidative activity. The active compounds in MPME act as natural remedies for diabetes treatment. However, further studies are required for isolating the active compounds and testing them in animal models of diabetes.

## Figures and Tables

**Figure 1 metabolites-12-00947-f001:**
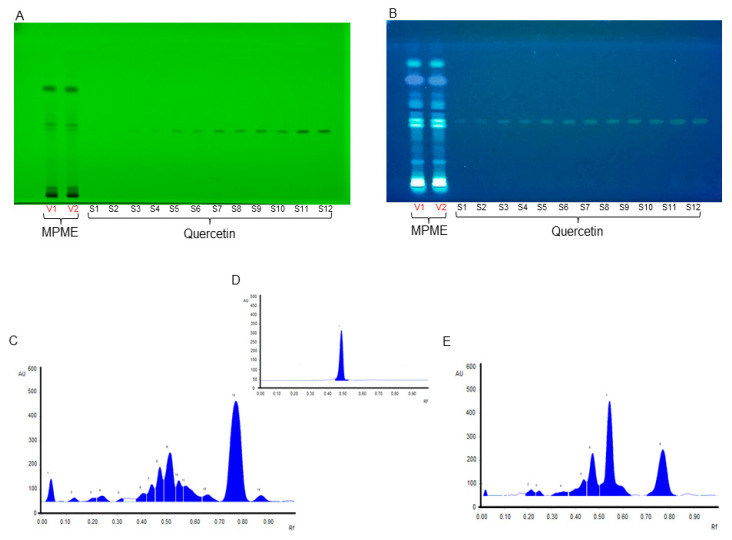
High-performance thin-layer chromatography (HPTLC) chromatoplate of methanolic heartwood extract of *Pterocarpus marsupium* (MPME) (V1-V2) and Quercetin (S1–S12) under 254 nm (**A**) and at 366 nm (**B**). HPTLC chromatogram of MPME scanned at UV 254 nm (**C**), HPTLC chromatogram of Quercetin scanned at UV 254 nm (**D**), and HPTLC chromatogram of MPME scanned at UV 366 nm (**E**).

**Figure 2 metabolites-12-00947-f002:**
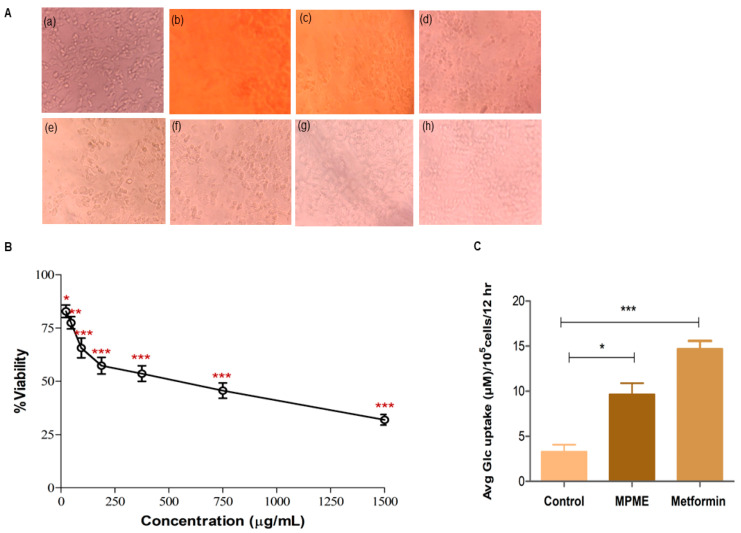
(**A**) Morphological changes in HepG2 cells after MPME treatment for 24 h. Untreated (Control) (**a**), Methanolic heartwood extract of *Pterocarpus marsupium* (MPME) (1500 µg/mL) (**b**), MPME (750 µg/mL) (**c**), MPME (375 µg/mL) (**d**), MPME (187.5 µg/mL) (**e**), MPME (93.75 µg/mL) (**f**), MPME (46.87 µg/mL) (**g**), MPME (23.43 µg/mL) (**h**). (**B**) Effect of MPME on HepG2 cell viability analyzed by 3-(4,5-dimethylthiazol-2-yl)2,5-diphenyl tetrazolium bromide (MTT) assay. Cells were exposed to increasing concentrations of MPME from 1500 µg/mL to 23.43 µg/mL for 24 h. Data are represented as means [±] standard deviation (SD) (n = 3). The asterisks (*) indicate statistical significant differences with the control (* *p* < 0.05, ** *p* < 0.01, *** *p* < 0.001, ns *p* ˃ 0.5). (**C**) Effect of MPME (93.75 µg/mL) on glucose uptake in HepG2 cells for 24 h. Data were analyzed by unpaired, two-tailed, *t*-test and presented as (±) standard error of mean (SEM) (n = 3) with significance * *p* < 0.05, *** *p* < 0.001 compared to untreated control.

**Figure 3 metabolites-12-00947-f003:**
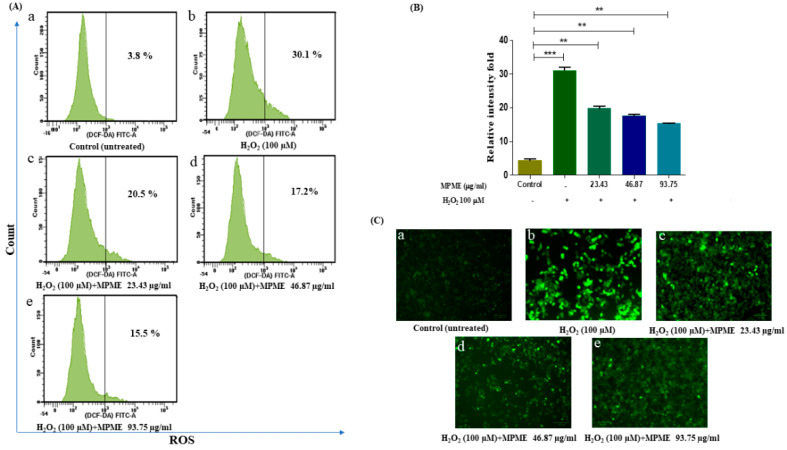
Flow cytometry analysis of intracellular reactive oxygen species (ROS) by H_2_O_2_-induced HepG2 cells, quantified by DCFH-DA probe. (**A**) Fluorescent intensity (ROS) of untreated and unstimulated HepG2 cells (control) was 3.8% (**a**) H_2_O_2_ induced cells was 30.1% (**b**) H_2_O_2_ induced cells + 23.43 µg/mL MPME was 20.5% (**c**) H_2_O_2_ induced cells + 46.87 µg/mL MPME was 17.2% (**d**) H_2_O_2_ induced cells + 93.75 µg/mL MPME was 15.5% (**e**) respectively. (**B**) Relative intensity fold of DCFDA staining. The asterisks (*) indicate statistical significant differences with the control (** *p* < 0.01, *** *p* < 0.001). + and − indicate the presence and absence of MPME/H_2_O_2_-100 μM (**C**) Fluorescence microscopy images (20×) of MPME stained with DCFDA to visualize ROS accumulation. In MPME treatments for 24 h, a decline in ROS generation compared to H_2_O_2_ stimulated cells was observed.

**Figure 4 metabolites-12-00947-f004:**
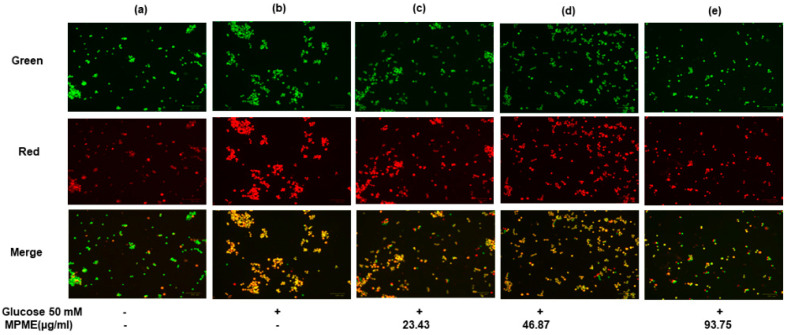
Effect of methanolic heartwood extract of *Pterocarpus marsupium* (MPME) on high glucose HG (50 mM)-induced apoptosis in HepG2 cells with acridine orange (AO)/EtBr staining. Fluorescence observation of HepG2 cells (unstimulated and untreated). Control (**a**); HG (50 mM) (**b**); High glucose (HG) + 23.43 µg/mL (**c**); HG + 46.87 µg/mL MPME (**d**); HG cells + 93.75 µg/mL MPME; (**e**) treated cells, with dual staining (AO (100 μg/mL): EtBr (100 μg/mL)) was in a 1:1 ratio. + and − indicate the presence and absence of MPME/HG (50mM). The HepG2 cells were examined under a fluorescence microscope in three different channels (green, red, and merged) with 20× magnification. The green nucleus of viable cells and the blood-red nucleus of apoptotic cells showed the morphological alterations of apoptosis following 24 h of drug treatment.

**Figure 5 metabolites-12-00947-f005:**
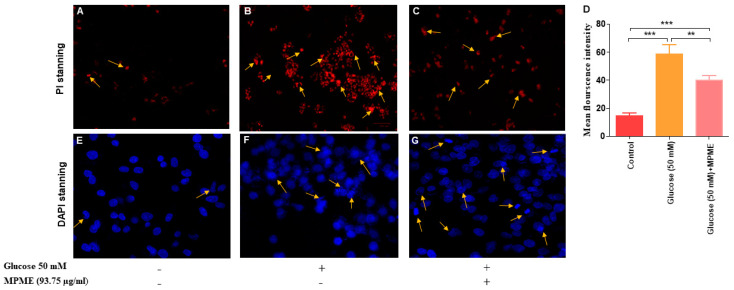
PI is a membrane-impermeant dye and stains only dead cells. Treatment of methanolic heartwood extract of *Pterocarpus marsupium* (MPME) 93.75 µg/mL for 24 h rescued the high glucose (HG)-induced HepG2 cells from apoptosis. PI stanning: normal control (**A**); HG (50 mM) (**B**); HG + 93.75 µg/mL MPME (**C**). Cell morphology was observed under a fluorescence microscope with 20× magnification. Yellow arrows signify the apoptotic cells. Fluorescence intensity after PI staining is represented by a bar diagram (**D**). Data are expressed as unpaired one-tailed *t*-test and are presented as the mean intensity of PI-stained cells. The asterisks (*) indicate statistical significant differences with the control (** *p* < 0.01, *** *p* < 0.001). DAPI Staining: confocal microscopic study untreated (control) (**E**); HG (50 mM) (**F**); HG + 93.75 µg/mL MPME (**G**). Cell morphology was observed under a confocal microscope at 40× magnification. Yellow arrows indicate apoptotic cells.

**Figure 6 metabolites-12-00947-f006:**
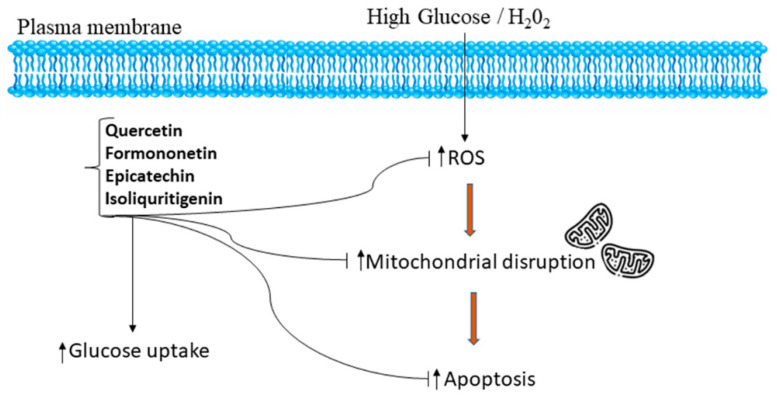
Schematic representation of the mechanism of potential antidiabetic and cellular antioxidant activities conferred by bioactive compounds such as epicatechin, quercetin, isoliquiritigenin, and formononetin identified by UPLC-MS and HPTLC analysis from a heartwood extract of *Pterocarpus marsupium*. The identified bioactive compounds relieve oxidative stress and attenuate apoptosis by regulation of ROS in a high glucose, H_2_O_2_-induced environment in HepG2 cells [48,49].

**Table 1 metabolites-12-00947-t001:** Phytocompounds identified in MPME by GC-MS analysis.

RT	Compound	Molecular Formula	Mol Weight g·mol^−1^	Peak Area
8.832	Benzaldehyde	C_7_H_6_O	106.12	19.92
10.153	Succinic acid	C_5_H_8_O_4_	10.153	0.76
11.120	Ethyl hydrogen succinate	C_6_H_10_O_4_	146.14	0.20
11.933	Cyclohexane	C_6_H_12_	84.16	0.11
14.388	5-Dimethoxybenzaldehyde	C_9_H_10_O_3_	166.17	0.98
14.691	Isoeugenol	C_10_H_12_O_2_	164.204	0.41
15.091	2,4-bis (1,1-dimethylethyl	C_17_H_3_OSi	278.5	15.89
15.669	Penta-fluoropropionic acid	C_3_HF_5_O_2_	164.03	1.66
15.944	2′,5′-Dimethoxypropiophenone	C_11_H_14_O_3_	194.23	0.04
16.722	Docosane	C_22_H_46_	310.6	0.12
17.701	Cyclopentane	C_5_H_10_	70.13	0.32
18.095	Myristic acid	C_14_H_28_O_2_	228	0.80
18.307	Oleyl alcohol	C_18_H_36_O	268.5	0.03
18.748	3,5-Dimethoxybenzyl alcohol	C_9_H_12_O_3_	168.19	2.62
19.463	1,2-Benzenedicarboxylic acid	C_24_H_38_O_4_	390	0.20
19.658	Pentadecanoic acid	C_15_H_30_O_2_	242.403	0.36
19.846	Pentatriacontane	C_35_H_72_	492.9	0.02
20.361	Citronellal	C_10_H_18_O	154.25	0.09
20.693	Isotridecanol	C_13_H_28_O	200.36	0.10
21.374	Dibutyl phthalate	C_16_H_22_O_4_	278	1.21
21.394	Phthalic acid (1,2-Benzenedicarboxylic acid)	C_8_H_6_O_4_	166.13	1.21
21.820	Hexadecanoic acid	C_16_H_32_O_2_	256	21.55
22.736	Oxiranepentanoic acid	C_5_H_10_O_2_	102.13	0.06
24.064	14b-pregnane	C_21_H_36_	288.5	0.02
24.207	Heptadecanoic acid	C_17_H_34_O_2_	270.5	0.14
26.530	Octadecanoic acid	C_18_H_3_6O_2_	284	20.01
25.792	9,12-Octadecadienoic acid	C_18_H_32_O_2_	280	0.33
30.449	Oleic Acid	C_18_H_34_O_2_	282.47	0.19

**Table 2 metabolites-12-00947-t002:** List of Phytoconstituents identified by ultra-performance liquid chromatography-mass spectroscopy(UP-LCMS) in methanolic heartwood extract of *Pterocarpus marsupium*.

RT	Tentative Mass	Exact Mass	Phytoconstituent	Formula	Class	Mass Record
3.047	179.0136	178.18700	2-Methoxycinnamic acid	C_10_H_10_O_3_	Cinnamic acid	Mass Bank ID (MBID):PR305807
3.047	291.1891	290.27100	Epicatechin	C_15_H_14_O_6_	Flavonoid	MBID:PS045605
3.047	453.1309	452.13190	Aspalathin	C_21_H_24_O_11_	Flavonoid	MBID:BS003543
3.047	463.2752	462.07983	Kaempferol-3-Glucuronide	C_21_H_18_O_12_	Flavonoid	MBID:PR101022
3.047	417.0264	416.4	Pterocarposide	C_21_H_20_O_9_	Flavonoid	Pub-Chm CID;01012651
3.047	433.0309	432.10565	Kaempferol-7-O-alpha-L-rhamnoside	C_21_H_20_O_10_	Coumarins	MBID:PR100942
3.047	434.8582	434.12131	Naringenin-7-O-glucoside	C_21_H_22_O_10_	Flavonoid	MBID:BS003367
3.047	437.2526	436.13696	Phloridzin	C_21_H_24_O_10_	Flavonoid	MBID:CE000071
3.047	437.2526	436.18860	Artocaprin	C_26_H_28_O_6_	Flavonoid	MBID:BS003597
4.000	237.0566	236.17763	Curcumol	C_15_H_24_O_2_	Sesquiterpenoid	MBID:FIO01064
4.000	257.0337	256.25699	Iso-liquiritigenin	C_15_H_12_O_4_	Hydroxychalcone	MBID:PR302707
4.000	257.0337	256.07355	Pinocembrin	C_15_H_12_O_4_	Flavonoid	MBID:BML00869
4.000	257.2227	256.30099	Trans-pterostilbene	C_16_H_16_O_3_	Stilbenoid	MBID:PR308326
4.000	485.1395	284.35550	Dihydroartemisinin	C_15_H_24_O_5_	Sesquiterpene	MBID:NGA00138
4.000	485.1395	284.06848	Wogonin	C_16_H_12_O_5_	Flavonoid	MBID:TY000033
4.000	285.1395	284.03207	Rhein	C_15_H_8_O_6_	Cassic acid	MBID:BML01013
4.000	485.1395	284.06848	Biochanin A	C_16_H_12_O_5_	Anthraquinone	MBID:PN000109
4.000	303.1256	302.04226	Quercetin	C_15_H_10_O_7_	Flavonoid	MBID:PN000111
4.000	337.2532	336.12357	Berberine	C_20_H_18_NO	Flavonoid	MBID:KO008886
4.000	336.1189	335.17331	Senecionine	C_18_H_25_NO_5_	Alkaloid	MBID:NA002260
4.000	352.2504	351.16818	Retrorsine	C_18_H_25_NO_6_	Alkaloid	MBID:BML82062
4.374	241.0898	240.25	Moscatin	C_15_H_12_O_3_	Phenol	Pub-Chm CID:194774
4.374	269.1331	268.26	Tectochrysin	C_16_H_12_O_4_	Flavonoid	Pub-Chm CID:5281954
4.374	268.9441	268.26	Formononetin	C_16_H_12_O_4_	Isoflavonoid	Pub-Chm CID:5280378
4.374	457.4785	456.53900	Vindoline	C_25_H_32_N_2_O_6_	Monoterpene indole alkaloid	MBID:PM000601
6.519	149.0768	148.05243	Citramalic acid	C_5_H_8_O_5_	Hydroxy-fatty acid	MBID:PR100770
6.519	240.0815	239.09464	6-Aminoflavanone	C_15_H_13_NO_2_	Flavonoid	MBID:JP000716
6.519	458.0455	457.15842	Amygdalin	C_20_H_27_NO_11_	Organooxygen compound	MBID:TY000092

## Data Availability

Data are contained within the article and Appendix A.

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
