# Peer review of "Heartwood Extract of Pterocarpus marsupium Roxb. Offers Defense against Oxyradicals and Improves Glucose Uptake in HepG2 Cells"

_metabolites, 2022, doi:10.3390/metabo12100947_

Round 1

Reviewer 1 Report

Dear Author, I noted that you presented many results showing that Heartwood extract of Pterocarpus marsupium (MPME), mainly its methanolic extract could be anti-diabetic. Their analysis and results are clear and useful for readers to understand their data. But some improvements could be made to clarify your manuscript.

In my "computational medicinal chemistry view", I missed some results directed to the main anti-diabetic targets despite the authors hypothesizing some mechanism based on their results (mainly in Fig. 6). Since the glucose-lowering level is a complex mechanism with many steps, the researchers could focus on one target and (theoretically) explore a few of them as NA+/Cl- reuptake protein, hexokinase, glucose-6-phosphorylate,... and other deposited on Protein Data Bank. Currently, many works employ docking methods to overcome experimental limitations (e.g., many targets) and choose the most probable interaction target. 

 Finally, the compounds described here are plentifully presented in literature and the authors could discuss in more detail if their roles are new or not. Hoping that my observation could improve this manuscript,

Regards.

Author Response

Comments and Suggestions for Authors

In my "computational medicinal chemistry view", I missed some results directed to the main anti-diabetic targets despite the authors hypothesizing some mechanism based on their results (mainly in Fig. 6). Since the glucose-lowering level is a complex mechanism with many steps, the researchers could focus on one target and (theoretically) explore a few of them as NA+/Cl- reuptake protein, hexokinase, glucose-6-phosphorylate, and other deposited on Protein Data Bank. Currently, many works employ docking methods to overcome experimental limitations (e.g., many targets) and choose the most probable interaction target. 

Response: Thank you for your point and suggestion. We have modified Figure 6 showing the mechanism of cellular glucose uptake in a high glucose environment and the possible therapeutic effect of plant extract in managing the toxicity of high glucose in a very simplified illustration according to the results.

Further, this study can be elaborated to understand the molecular biology of the individual compounds responsible for the antidiabetic activity and moreover, the isolation of many metabolites such as Quercetin can be used to develop phytopharmaceutical to manage diabetes.  

Comment

 Finally, the compounds described here are plentifully presented in literature and the authors could discuss in more detail if their roles are new or not. Hoping that my observation could improve this manuscript

Response: The role of important compounds has been discussed in the manuscript as per the suggestion

Reviewer 2 Report

I have gone through the manuscript entitle; “Heartwood extract of Pterocarpus marsupium Roxb. offers defense against oxyradicals and improves glucose uptake in HepG2 cells.” which describes the use of heartwood extract for diabetic. The extensive work has been done and its good contribution toward to applied work for development of diabetic inhibitors. I recommended it for publication However I suggest to improve the following point (see in attachment of manuscript pdf) before final acceptance.

Author Response

Comments and Suggestions for Authors

I have gone through the manuscript entitle; “Heartwood extract of Pterocarpus marsupium Roxb. offers defense against oxyradicals and improves glucose uptake in HepG2 cells.” which describes the use of heartwood extract for diabetic. The extensive work has been done and its good contribution toward to applied work for development of diabetic inhibitors. I recommended it for publication However I suggest to improve the following point (see in attachment of manuscript pdf) before final acceptance.

Response: We thank you for the valuable suggestions and the highlighted points have been improved in the manuscript as per the suggestions

Reviewer 3 Report

In the paper entitled “Heartwood extract of Pterocarpus marsupium Roxb. offers defense against oxyradicals and improves glucose uptake in HepG2 cells”, the authors have reported results related to the evaluation of the potential biological activity of extract from Pterocarpus wood.

In general, the topic is very interesting because in many cases has been found extracts from natural sources that have biological activities. In general, the manuscript is well done and presented, and the results are sound.

I think that the manuscript needs to be modify by following these indications:

-          Page 4, paragraph 2.8.1, line 176; please specify the final concentration of the extracts used in the experiments;

-          Page 16, lines 546-549; I didn’t understand this sentence;

-          As general consideration about the Discussion I think that the concentrations at which MPME shows activity are high, ranging around hundreds microg/ml. The authors have considered that among the components of the MPME mix ther are some of there that are in conflict with the others? Will be possible purify and test the main components of the extract?

-          The Discussion in some parts could be reduced because there are sentences already reported in other parts of the manuscript.

In conclusion, in my opinion the manuscript needs of minor revisions before publication.

Best Regards

Author Response

Comments and Suggestions for Authors

In general, the topic is very interesting because in many cases has been found extracts from natural sources that have biological activities. In general, the manuscript is well done and presented, and the results are sound.

Comment

Page 4, paragraph 2.8.1, line 176; please specify the final concentration of the extracts used in the experiments;

Response: The final concentrations have been specifically mentioned in the manuscript as per the suggestion.

Comment

Page 16, lines 546-549; I didn’t understand this sentence.

Response: The sentence has been revised for better clarity.

Comment

As general consideration about the Discussion I think that the concentrations at which MPME shows activity are high, ranging around hundreds microg/ml. The authors have considered that among the components of the MPME mix ther are some of there that are in conflict with the others? Will be possible purify and test the main components of the extract?

Response: The concentration of MPME showing high activity is 93.75 µg/ml and 46.87 µg/ml and the given concentrations showed the least toxicity on HepG2 cells and further safety evaluations, and activity can be determined using in-vivo models. The current study tentatively identified many antidiabetic phytoconstituents and shows the quantification of Quercetin using HPTLC the presence of compounds showed no conflict with others. The study can be further elaborated to isolate the quercetin or other identified antidiabetic phytoconstituents and can be evaluated for antidiabetic activity using in-vivo models.

Comment

The Discussion in some parts could be reduced because there are sentences already reported in other parts of the manuscript.

Response: Thank you for your valuable suggestion, the repetitive sentences have been rectified in the manuscript according to the suggestion